# Ethics of End of Life Decisions in Pediatrics: A Narrative Review of the Roles of Caregivers, Shared Decision-Making, and Patient Centered Values

**DOI:** 10.3390/bs8050042

**Published:** 2018-04-26

**Authors:** Jonathan D. Santoro, Mariko Bennett

**Affiliations:** 1Division of Child Neurology, Lucile Packard Children’s Hospital at Stanford, Palo Alto, CA 94305, USA; 2School of Medicine, Stanford University, Stanford, CA 94305, USA; mlhowe@stanford.edu

**Keywords:** end of life, pediatrics, palliative care, shared decision-making

## Abstract

*Background:* This manuscript reviews unique aspects of end of life decision-making in pediatrics. *Methods:* A narrative literature review of pediatric end of life issues was performed in the English language. *Results:* While a paternalistic approach is typically applied to children with life-limiting medical prognoses, the cognitive, language, and physical variability in this patient population is wide and worthy of review. In end of life discussions in pediatrics, the consideration of a child’s input is often not reviewed in depth, although a shared decision-making model is ideal for use, even for children with presumed limitations due to age. This narrative review of end of life decision-making in pediatric care explores nomenclature, the introduction of the concept of death, relevant historical studies, limitations to the shared decision-making model, the current state of end of life autonomy in pediatrics, and future directions and needs. Although progress is being made toward a more uniform and standardized approach to care, few non-institutional protocols exist. Complicating factors in the lack of guidelines include the unique facets of pediatric end of life care, including physical age, paternalism, the cognitive and language capacity of patients, subconscious influencers of parents, and normative values of death in pediatrics. *Conclusions:* Although there have been strides in end of life decision-making in pediatrics, further investigation and research is needed in this field.

## 1. Introduction

A natural response in medical decision-making for a pediatric patient is to delegate authority to the supervising parent or caregiver. This frequently occurs while providing families with the specifics of developments, treatments, and prognosis associated with a child’s condition. In this setting, a pediatric patient’s input on medical decision-making can often be secondary, if considered at all. This is especially notable in end of life care and is certainly true for children born with limited cognitive, language, or motor function, or those unable to achieve full mental and/or physical capacity [1]. Yet each stakeholder contributes unique emotions, experiences, cultural beliefs, and medical acumen to the discussion of end of life care decisions. Thus the physician, as the frequent de facto moderator of the discussion, must not only make the most medically appropriate decisions but also understand very unique sets of psychology in promoting “pro-child” choices. 

Given the broad topic of pediatric end of life issues and the heterogeneity of conditions, cognitive levels encountered, and influencing factors in care, a traditional narrative review was deemed the most appropriate survey for the subject matter. A non-systematic critical analysis of both historic and up-to-date literature on the topic of end of life decision-making in pediatric care was performed. It explores nomenclature, the introduction of the concept of death, relevant historical studies, the current state of end of life autonomy in pediatrics, and future directions and needs.

## 2. The Death Concept and Initial Communications

The formation of a “death concept” is constructed differently at various points in a person’s lifecycle, and can influence the patient’s, physician’s, and parent’s understanding of death and dying [1]. Thus, the perceived experience and understanding of one’s mortality is often limited by the ability to have conscious thought about it in a form that is communicable to others. Adults frequently utilize vernacular changes that invoke willpower in this process, often opting to “fight” a disease, which is not observed with the same frequency in pediatric populations [1]. This cognitive framework is often used interchangeably with both life threatening illnesses, such as cancer (where patients may die, but this is not an inevitable outcome of the disease), and life limiting illnesses (where patients will die from the disease), although the latter will be the focus of this manuscript. 

Piaget theorized that all children move through discrete stages of sophisticated thinking and even younger children can posses adult-like cognitive skills that allow them to actively participate in meaningful medical decisions [2]. Critics advocate that while children may have adult-like cognitive abilities, their expressive communication skills may limit their ability to participate meaningfully in end of life discussions and decisions. Nonetheless, the issue is not whether children are less sophisticated problem-solvers than adults, but rather, how the quality and speed of their cognitive process influences the way an individual understands his/her circumstances. Patients of any age can experience this development in cognition and meaningfully evaluate environments, relationships, and circumstances they find essential to their quality of life. 

In pediatric populations, books and visual aids have often been used to introduce patients to terminology that they may lack in order to enhance standardized means of communicating preferences successfully [3,4]. Furthermore, the use of standardized communication allows for optimized information sharing, which is a prerequisite in the shared decision-making model [5]. Prior literature has demonstrated reduced rates of anxiety and depression in pediatric patients with life limiting illnesses who are able to utilize open forms of communication [6]. Accordingly, while communication is key for both the reduction of fear and ability to express preferences, adaptive methods that are age appropriate for the patient must be employed. This is extraordinarily pertinent to avoid misinterpretation of a child’s preferences regarding end of life care [1].

## 3. Shared Decision-Making and Its Limitations

The American Academy of Pediatrics (AAP) advocates that physicians involve patients in their treatment decisions in a “developmentally-appropriate” manner. This is commonly referred to as “shared” decision-making and is generally encouraged in pediatric care, both inside and outside of end of life issues. Yet, while desirable, limitations from both an age and cognitive reasoning perspective can make this model difficult to utilize.

Adolescents are the closest in age to adults yet, the shared decision-making model can be challenging for this population. Physicians and parents alike must carefully weigh the patient’s ability to use their possibly underdeveloped executive and impulse control in the context of their socio-emotional decision-making [7]. Many states in the United States of America already allow teens to self-consent for treatment or testing for “adult-like” conditions, such as sexually transmitted diseases, substance abuse, or pregnancy. A critical aspect of extending the decision-making process to adolescent minors in the aforementioned conditions is a thorough informed consent process at the cognitive level of the patient [7]. Specifically, increased review of long-term effects and consequences of treatment refusal or delay are critical, as adolescents often operate in a more limited time context, especially during times of stress [8,9]. Similarly, information and informed consent must be equally applied in a “developmentally-appropriate” manner when engaging in end of life decision-making in adolescent minors. 

In any decision-making process, adults are invariably influenced by cultural, societal, religious, and familial factors unique to the individual. When considering the cognitive maturity of a child, one must examine their environmental, social, media, and social-media influences, which can weigh even heavier on the formation of a child’s cognitive development and thought schema than that of their parents or familial upbringing [5]. A study by Weithorn, designed to analyze the decision-making limitations imposed by chronological age, found that nine-year-olds exhibit the ability to weigh important pieces of information when presented with medical scenarios [10]. Additionally, a critically ill patient is more often visited by adult family members, medical professionals, and a community of sound intelligence that influence a child’s ability to form cognitive assessments and develop with more thoughtfulness and maturity [5,10]. Interestingly, Coyne et al. reported that, in a small cohort of pediatric patients with life limiting illness, some patients preferred to not be actively involved in the decision-making process, but rather they desired and valued receiving information, voicing their preferences, and choosing how treatments were administered to them [11]. Accordingly, establishing a patient’s desired level of involvement in decision-making early in the end of life process is needed prior to determining communication and preference structure.

Obviously, there are particular situations in which a shared decision-making model is not applicable. Most notable in the field of pediatrics is the issue of non-accidental trauma, wherein a parent or caregiver is suspected of abuse. Prior studies have analyzed the laws and regulations around this issue and have concluded that the dramatic conflicts of interest in opting for non life-sustaining measures for children abused by parents is high and leads to deferred end of life decision-making due to the potential of legal repercussions [12]. To avoid conflicts of interest that may cause harm or prolong suffering, medical teams must be willing to utilize clinical ethics consultation and the local legal system early in the hospital course to act in the best interests of the child.

## 4. End of Life Care in Patients with Limited Capacity

The majority of life limiting pediatric cases occur before one year of life, making shared decision-making models impractical. Until recently, only one systematic protocol existed for dealing with terminal pediatric patients and it was specific to the surgical repair of meningomyelocele as a futility treatment [13]. In 2002, a group from the Netherlands developed the “Groningen Protocol” as a systemized approach to dealing with pediatric euthanasia issues in the neonatal period [14]. This protocol emphasized five criteria that should be used to determine whether euthanasia, or in many cases withdrawal of care, is feasible: The infant must have certain diagnosis or prognosis.He or she must have hopeless and unbearable suffering.Criteria 1 and 2 must be confirmed by at least one independent doctor.Both parents must give informed consent.The procedure must be carried out in accordance with acceptable medical standards.

The aim of this protocol was to minimize the suffering of pediatric patients born with “hopeless” diseases, or prognoses where the quality of mental and physical life would be severely impaired or where death was imminent. Examples from the Groningen protocol included lung and kidney hypoplasia, Trisomy 13, extreme hypoxemia at birth, and severe brain abnormalities, such as anencephaly. This protocol has received heavy criticism both domestically and abroad, but was revolutionary in its purpose of providing a systemized approach to difficult end of life decisions in neonates. 

The Groningen protocol places the physician in a role of discussion leader or moderator, a framework for conversation while still addressing the importance of “parental consent” [14]. This protocol shifted the perceived framework of decision-making from a sole parental consideration toward consensus between medical caregivers and parents, an approach beneficial for overcoming religious and cultural issues encountered in end of life and general medical care [15]. Despite the criticism, the intention of the Groningen Protocol was not to “kill babies”, but rather to prioritize quality of life of patients lacking autonomy over the abilities of technology to prolong life [16].

Since this seminal report, multiple international teams have provided framework-based recommendations on end of life care and palliation in the very young. Recommendations from France [17], the Netherlands [18], and Switzerland [19] have provided the basis for further query into standardized approaches internationally. Similarly, the AAP has recently taken action in providing 12 consensus-based guidelines for its members and continues to advocate for additional research on quality measures in this area [20,21,22].

Although distinctly different, pediatric patients with diminished cognitive, motor, or language capacities similarly lack the ability to participate in shared decision-making regarding end of life issues. In spite of this, there are no large-scale pediatric studies to date addressing this important topic. The majority of end of life decision-making comes from adult literature in dementia and other neurodegenerative diseases, although the bulk of these studies have focused on the ability of proxies to make decisions based on prior patient preferences [23,24,25]. The ethical dilemma of ensuring patient preference in a child who has never achieved the ability to communicate, or has limited capacity to do so, is complex. In these situations, acting in the best interest of the child and avoiding prolonged suffering should be the primary goal of all decision-makers and the path facilitated by physicians. These cases are most appropriate for in hospital ethics consultation, which is addressed later in this manuscript.

## 5. Beyond Shared Decision-Making—Parental Authority and Conflicts

The role of parents in pediatric end of life care is paramount, but also one fraught with difficulty in situations where grieving may influence typically clear decision-making abilities [26]. The primary obligation of a parent is to keep their children healthy, protected, and safe—a responsibility that is accentuated when a child has a life limiting or threatening medical condition. Inherent to this parental obligation is a neuro-endocrine shift that can lead parents to become narrow-minded about what is best for their child [27]. Applying a broad standard under which parents are responsible and able to maximize their child’s well-being may be too optimistic when parents are confronted with life-altering changes for their child [1]. This protective parental role, while critically important and valid, must be balanced and possibly tempered with sound medical practice that weighs quality of life and realistic expectations of outcomes. This is of even more importance in patients who lack the capacity to communicate their goals and wishes regarding their medical care. It is important to involve parents in regard to discussing and educating them on the child’s development and condition. However, a parent’s relationship with her child may default to survivalist and an individual’s relationship to himself will vary independently [27,28].

The preferences parents have for their child’s end of life care may differ from the patient’s. Most studies indicate that adults prefer to spend their last days at home [29,30,31], but it is unclear whether these preferences, as surrogate decision-makers, apply to their children [32]. An understanding of such desires and intentions is important, as a majority of children with chronic or perinatal/genetic disorders die in a hospital setting. Parents may also base their decisions on cultural, moralistic, or other preferences unknown to the care team. Several studies in pediatric patients have identified that religion and spiritual beliefs largely influence decision-making in end of life care [33,34,35]. In addition, identification as “highly religious” has been previously linked to delays in planning for end of life care and sustained intervention deemed not beneficial to patients [34,35].

Decision-making preferences amongst parents with critically ill children have been analyzed previously, with the majority of work being undertaken in neonatology and children with complex congenital heart disease. Factors that have been recognized to effect likelihood of aggressive life sustaining measures include higher education level [36], ethnicity [35,37], and lack of prior exposure to death of a relative [38]. Other factors, such as maternal age, maternal gravida, maternal parity, and socioeconomic status of parents, have been studied with inconclusive results [39]. The mindful shifts necessary to act objectively can be difficult for parents, especially as many subconscious biases and influences can alter the decision-making process [35,36,37,38,39]. While these interventions were traditionally handled by medical teams, primary care physicians, and families, the last two decades have been marked by the development of standardized clinical ethics consultations and the advance of palliative care as a medical subspecialty, allowing medical teams to more appropriately handle end of life issues in this unique population.

## 6. Ethics Consultation and the New Era of End of Life Care

End of life decision-making in pediatric care is complex, and very frequently heterogeneous, in the factors associated with individual cases. The rise in the availability and presence of ethics consultations at many tertiary and quaternary pediatric care centers over the past few decades has changed the landscape for medical teams encountering difficulty with end of life issues [40,41]. Traditional barriers to utilization of clinical ethics consults included lack of knowledge of such committees, fear of utilization, belief that decision-making must come from the primary care giver (including fear of other physicians or nurses interacting with families), delays in decision-making, and lack of formal support [42]. Although each hospital system and region is different in regard to the issues surrounding ethical consultation, knowledge of committees and diminished stigma around utilization are important universal interventions that may benefit end of life decision-making in complex pediatric patients [42,43]. For some patients, however, utilization of a hospital ethics committee is not necessary when medical teams and parents or guardians are in agreement about the patient’s care goals, and in these situations, a focus on symptom management and comfort is prioritized.

Clinical ethics committees have undoubtedly assisted medical teams and families with end of life decision-making in patients with life limiting illnesses, although the concurrent development of palliative care in pediatrics has also extended the timeline of and approach to end of life issues of patients with life threatening diseases. Palliative care is an approach of diminishing suffering in persons with life limiting or life threatening diseases, although dealing with acute end of life issues is not always necessary, especially initially. Both life threatening and life limiting diseases benefit from palliative care approaches and early involvement is preferred as life threatening diseases, such as cancer, have the potential to become life limiting in situations where treatments are ineffective or exhausted.

The overlap between clinical ethics and palliative care is substantial, although the ultimate function of palliative care, as previously addressed, is more traditionally focused on symptom management (often pain-related) in the context of life threatening and/or life limiting conditions [44,45]. Palliative care specialists can work in conjunction with clinical ethics teams and medical teams to provide a multi-disciplinary approach to end of life decision-making in patients with life limiting diseases, with a primary focus on comfort [44]. These teams are most often involved longitudinally, which provides the added ability to reassess interventions as goals of care change [46]. The added value of palliative care is notable for patient, family, and caregiver teams and should be a routine part of end of life care, whether or not clinical ethics consultation is needed [47].

In practice, a community-wide commitment to increasing early access to palliative services for terminal pediatric oncology patients has had positive impacts: reduction of patient and parental suffering [48]. These effects are attributed to improved family’s understanding of their child’s prognosis, earlier and improved dialogue leading to advanced directives or outlining of child/parent preferences, and utilization of shared decision-making [48,49]. The introduction of palliative care as early as diagnosis, even in non-terminal oncology cases, is also linked to better outcomes [50]. Screening tools in a pediatric intensive care unit (ICU) settings increase access to palliative care, but risk increasing demand, which may lead to a shortage of services [51]. Importantly, as we learn to better utilize and integrate palliative care services into practice, we must not delegate our responsibility for shared decision-making and patient advocacy strictly to these consultant services. 

## 7. The Primary Caregiver and Non-Consultant Based Intervention

Parents and physicians may be thrust into the forefront of decision-making in pediatric end of life care, but it is not necessarily a role relished by either. Interestingly, 56% of parents in one study believed they made the ultimate decision in their child’s care-plan; of those who did not, only 7% later wished that they had [52]. This study, and others, highlights that an individual parent’s desire to be the ultimate decision-maker varies. Beyond their role as medical providers, physicians have an obligation to broker or participate in the decision-making process; these partnerships between parents and the care team are critically important [53]. The PELICAN consortium studied the experiences of Swiss families who had lost a child to illness, and found that while families reported an overall positive experience in their interactions and share decision-making with the healthcare team, their satisfaction varied largely by specialty. One-third of parents whose child faced a resuscitation decision felt they made that choice alone, while most families (52%) who interacted with neurologists as their primary caregivers felt they made the decision alone [52]. The importance of highlighting these differences lies with intentionality: does physician guidance lead to the best outcomes and the least suffering for pediatric patients at the end of life? What are the barriers to shared decision-making involving the physician? 

## 8. Conclusions

The intent of invoking end of life decision-making is often of pure intent, but with the myriad of complex social, personal, and medical factors overlaying the decision-making process. Routine and standardized involvement of health care providers with training in clinical ethics and palliative care may be the best way for physicians to navigate this arena in pediatrics and provide the most appropriate care for each individual patient. Further investigation into this new and burgeoning field is necessary.

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
