# Peer review of "Ethics of End of Life Decisions in Pediatrics: A Narrative Review of the Roles of Caregivers, Shared Decision-Making, and Patient Centered Values"

_behavsci, 2018, doi:10.3390/bs8050042_

Round 1

Reviewer 1 Report

while interesting, your paper would be significantly improved by addr swing issues with mature minors facing a terminal illness. Additionally, legal aspects of child injury should be touched on. For example, the brain death in an infant whose parent may be charge with assault will be changed to manslaughter as the infant is removed from life support. While this should not stop the medical team acting in the best interest of the child, such a case might be best served with the hospital ethics team in involved. The routine use of the hospital ethics team is not mentioned. 

Author Response

March 21, 2018

Dear Reviewer #1:

            Thank you for taking the time to review this manuscript, we sincerely appreciate your comments and hope that you find the attached resubmission worthy of publication. We are very appreciative of the targeted comments provided in your review and believe that in addressing them we have made this manuscript significantly more readable and informative to the readership of Behavioral Sciences. Regarding your specific comments:

1)    Addressing swing issues and problems with mature minors facing terminal illness

The authors are in agreement that this was one of the areas that was lacking in the original submission of the manuscript. We have now added two expanded regions to the manuscript to address this issue. Firstly we provided a significant expansion of adolescent (or mature minor issues) in the newly revised section entitled: Shared decision-making and its limitations. Here we provide useful resources on how teens are more likely to make decisions with treatment refusal and delay in end of life care and that this can be attributed to neurocognitive alterations in the developing teen brain (reference 7). Specifically, we have now cited references that address both this specific issue and how decision-making processes are affected during times of stress (references 8 and 9). Additionally, we also expanded this to include patients, even mature adolescents/minors, who wish to not be as actively involved in the decision making process but how information sharing and opinion forming is still critical for the shared decision making model (reference 11).

2)    Addressing legal aspects of child injury

The authors are in complete agreement with the reviewer that this topic is of particular interest in a pediatric-based review of end of life issues. The authors have included additional information and references in the newly revised section entitled: Shared decision-making and its limitations. Specifically, the authors highlight the dramatic conflicts of interest in parental decision making in the setting of non-accidental trauma and cited a medico-legal review manuscript from Pediatrics (reference 12) that highlights key issues to this problem. Further the authors support early and close involvement of the both the hospital clinical ethics team and the local legal systems to optimize care for the patient and to avoid suffering or extraneously prolonged life when these situations arise.

3)    Routine use of the hospital ethics committee

The authors are in agreement with the reviewer that this was a problematic omission from the first draft of the manuscript. The authors have now included a dramatic expansion of the manuscript to address clinical ethics consultation and palliative care in the newly devised section entitled: clinical ethics consultation and palliative care teams. This section now provides vastly improved information about the roles (as well as some history) of these two unique teams (references 40, 41, 44, and 45), their overlap and barriers to use (references 42 and 43), and how they should most optimally be used (references 46 and 47).

Most Sincerely,

Jonathan D. Santoro, MD

Reviewer 2 Report

Thank you for submitting this review. 

I was excited when I read the title but felt a bit disappointed when I had finished reading the paper. The Review did not follow any framework that I could see and it was unclear why such headings arose from. I realise this is not research but does require a rigorous procedure and such an approach is not transparent here.

I realise the paper was written in America but as this is an international Journal I think the terminology adopted could have been more appropriate for the readership. Children with limited faculties, terminal care, the interchangeable use of EOL and palliative care without explanation. 

Many of the references were older and there was no mention of the work done by Imelda Coyne did in relation to decision making in children. 

I think this needs major reworking before it reaches publishable international standard. I agree though it is a pertinent topic which needs exploring.

Author Response

March 21, 2018

Dear Reviewer #2:

            Thank you for taking the time to review this manuscript, we sincerely appreciate your comments and hope that you find the attached resubmission worthy of publication. We are very appreciative of the targeted comments provided in your review and believe that in addressing them we have made this manuscript significantly more readable and informative to the readership of Behavioral Sciences. Regarding your specific comments:

1)    Manuscript Framework

The authors are in agreement that the overall framework of the original manuscript was unclear and vague. The authors have attempted to remedy this by 1) introducing the nature of the review in the introduction and abstract; 2) providing a rational for section break down in the last paragraph of the introduction; and 3) revising the titles of sections to improve fluidity and make the manuscript more encompassing. With regards to the latter, the sections of this manuscript now proceed as follows:

-       Introduction

o   Now includes type of study and summary of sections in a more fluid manner

-       The Death Concept and Initial Communications

o   Focusing on introducing the new language of end of life to children and broadening to include why this is done, how this can be done, and alternative modalities utilized in children (specifically books and visual aids). We have also enhanced this section with references that demonstrate open communication decreases anxiety and depression in patients and also allows for the ability to fully participate in the shared decision-making model (references 1, 4, 5, and 6).

-       Shared Decision-Making and its Limitations

o   This new section introduces the idea of shared decision-making and how this concept is promoted (in the most age appropriate manner) by the American Academy of Pediatrics.

o   The authors present limitations to this model in adolescents (references 7,8 and 9) with regards to difficulties with assessment of long term risk and increased rates of treatment refusal or delay, age related limitations (references 5, 9 and 10), lack of desire to be involved (reference 11 by Imelda Coyne) and cases of non-accidental trauma (reference 12)

-       End of Life Care in Patients with Limited Capacity

o   This section is in many ways similar to the original section as the introduction of the Groningen protocol was of importance for historical introduction. This was then supplemented by national studies from France, the Netherlands, and Switzerland (as well as the United States) on nationwide frameworks that have come from this (references 17-22).

o   The above issues primarily related to neonatal conditions and there is an unfortunate paucity of information on patients with cognitive, language, or motor impairments and end of life decision-making. The authors chose to site end of life issues in older patients with dementia and other neurocognitive issues as a reference point although cited the need for further research and ethics consultations in these unique cases.

-       Beyond Shared Decision-Making – Parental Authority and Conflicts

o   This section identified the often conflicting goals of primary care giver and acting in the best interest of the child. The authors chose to highlight some of the previously established influencers (conscious and subconscious) on end of life care decisions which we believe adds an additional layer of paucity to having utilizing the traditional “paternalistic” approach to end of life issues (references 33-39).

-       Clinical Ethics Consultation and Palliative Care Teams

o   This section was unfortunately omitted from the original manuscript but we believe that this expanded section provides excellent context and “where are we going” information for the reader. This section now highlights some of the historical points of ethics consults and palliative care teams as well as primary goals of each and overlap between the two. Additionally, the authors identified barriers to their full utilization as an additional piece of information as these issues appear to be universal.

-       The Primary Caregiver and Non-Consultant Based Intervention

o   This section wraps up our findings and identifies some of the common parental issues encountered in end of life care. This section focuses on “after the fact” analysis of the entire end of life process and provides an interesting context to end the piece on.

-       Conclusions

2)    Standardization of Terminology

The authors are in agreement about the non-standardized terminology utilized in the original manuscript. The following production has attempted to utilize more broadly used international language. Additionally the authors have clarified end of life and palliative care more clearly as this was used interchangeably at a few points in the original manuscript.

3)    References

The authors are in agreement with the reviewer and have dramatically expanded the references in this submission. The authors have more than doubled the amount of references in this resubmission from 22 to 53. The authors felt that a combination of older pieces would be worthwhile to contrast on some of the more up to date studies on emerging issues in pediatric end of life care. The authors also have taken the time to add citations for several more recent position statements from the American Academy of Pediatrics over the last decade to the piece for a more systematic approach to documenting the emerging and changing opinions about end of life care in pediatric medicine. Finally, as the reviewer most aptly noted, there was a void of references by Imelda Coyne who has published quite a bit on this topic and are now included in the manuscript (references 4, 5, and 11).

Most Sincerely,

Jonathan D. Santoro, MD

Round 2

Reviewer 2 Report

Thank you for resubmitting this paper and for systematically addresses the issues raised in the original feedback.

I think the paper is much stronger. If this is a narrative review I think it should be indicated in the title. Also I think you should in a sentence in the introduction discuss what a narrative review is and justify the merit of using it here.

I still think there is not absolutely clarity about how palliative care and end of life care fit together ie definitions.Also cancer is life threatening but can become life limiting if treatment options exhausted. 

The review has benefited from much more literature.

I think if the above were addressed it could be publishable.

Thank you and best wishes

Author Response

April 4th2018

Dear Reviewer 1:

Thank you for taking the time to review this manuscript, we sincerely appreciate your comments and hope that you find the attached resubmission worthy of publication. We are very appreciative of the targeted comments provided in your review and believe that in addressing them we have made this manuscript significantly more readable and informative to the readership of Behavioral Sciences. Regarding your specific comments:

1.     Narrative review identification in title.

The authors are in agreement with the reviewer with regards to this concern and have now changed the title to be more reflective of the nature of the manuscript as follows: Ethics of end of life decisions in pediatrics: a narrative review of the roles of caregivers, shared decision-making, and patient centered values”

2.     Justification and identification of narrative review in the introduction.

The authors are again in agreement with the reviewer on this issue. We have now identified earlier in the manuscript that we are using a narrative review and provide justification for its use based on the information and broad nature of the topic. The last paragraph of the introduction now reads: 

“Given the broad topic of pediatric end of life issues and the heterogeneity of conditions, cognitive levels encountered, and influencing factors in care, a traditional narrative review was deemed the most appropriate survey for the subject matter. A non-systematic critical analysis of both historic and up-to-date literature on the topic of end of life decision-making in pediatric care was performed and explores nomenclature, the introduction of the concept of death, relevant historical studies, the current state of end of life autonomy in pediatrics, and future directions and needs.”.

We believe this clarification makes the piece much easier to approach now and thank the reviewer for this insight.

3.     Clarification of the differences between palliative care and end of life care.

Thank you for this excellent feedback. We agree that this could be clearer and have added an additional paragraph in the Ethics Consultation and the New Era of End of Life Care section of the manuscript. This paragraph reads: 

Clinical ethics committees have undoubtedly assisted physicians, medical teams, and families with end of life decision-making in patients with life limiting illness although the concurrent development of palliative care in pediatrics has also extended the timeline and approach to end of life issues to patients with life threatening diseases as well. Palliative care is an approach of diminishing suffering in persons with life limiting or life-threatening diseases although dealing with acute end of life issues is not always necessary, especially initially. Both life threatening and life limiting diseases benefit from palliative care approaches and early involvement is preferred as life threatening diseases, such as cancer, have the potential to become life limiting in situations where treatments are ineffective or exhausted.”. 

As referenced above, the authors have also provided more clear definitions of life threatening and life limiting disease earlier in the manuscript which serves as a more clear differentiator for the roles of palliative and end of life care as well.

4.     Clarification on how diseases like cancer are life threatening but can become life limiting when treatment options are exhausted.

The authors are in agreement that this should be more cleanly and succinctly defined as life threatening and life limiting conditions are in fact different. The authors have now introduced a more clear early introduction to the terms in the 2ndsection of the manuscript (The Death Concept and Initial Communications) in the first paragraph as follows: 

This cognitive framework is often used interchangeably with both life threatening illnesses such as cancer (where patients may die but this is not an inevitable outcome of the disease) and life limiting illness (where patients will die from a disease) although the later will be the focus on this manuscript.”

The authors have also added on a further statement to the end of the clarification of differences between palliative and end of life care (as above) as the definitions of life limiting and life threatening conditions is relevant for the differentiation of these two topics: 

Both life threatening and life limiting diseases benefit from palliative care approaches and early involvement is preferred as life threatening diseases, such as cancer, have the potential to become life limiting in situations where treatments are ineffective or exhausted.”

This is further expanded again for consistency in the following paragraph. Regarding palliative care:

These teams are most often involved longitudinally, which provides the added ability to reassess care as time and goals of care change, especially in patients diagnosed initially with life threatening disease where treatment is ineffectual or exhausted leading to life limiting disease states [46].”

The authors sincerely appreciate the reviewers comments and hope that these edits in addition to other minor edits for clarification throughout the manuscript will improve the quality of this manuscript.

Most Sincerely,

Jonathan D. Santoro, MD
